# Obsessive-Compulsive Disorder in PANS/PANDAS in Children: In Search of a Qualified Treatment—A Systematic Review and Metanalysis

**DOI:** 10.3390/children9020155

**Published:** 2022-01-26

**Authors:** Salvatore Cocuzza, Antonino Maniaci, Ignazio La Mantia, Francesco Nocera, Daniela Caruso, Sebastiano Caruso, Giannicola Iannella, Claudio Vicini, Elio Privitera, Jerome Rene Lechien, Piero Pavone

**Affiliations:** 1Department of Medical and Surgical Sciences and Advanced Technologies “GF Ingrassia”, ENT Section, University of Catania, 95123 Catania, Italy; s.cocuzza@unict.it (S.C.); antonino.maniaci@phd.unict.it (A.M.); ignazio.lamantia@unict.it (I.L.M.); ciccionocera94@gmail.com (F.N.); sebastiano.caruso2404@gmail.com (S.C.); priviteraelio@gmail.com (E.P.); 2Unit of Clinical Pediatrics, A.O.U. “Policlinico”, P.O. “G. Rodolico”, University of Catania, 95123 Catania, Italy; danicaruso23@outlook.it; 3Department of Head-Neck Surgery, Otolaryngology, Head-Neck and Oral Surgery Unit, Morgagni Pierantoni Hospital, 47121 Forlì, Italy; giannicola.iannella@uniroma1.it (G.I.); claudio@claudiovicini.com (C.V.); 4Department of Sensory Organs, Sapienza University of Rome, 00194 Rome, Italy; 5Department of Human Anatomy and Experimental Oncology, Faculty of Medicine, UMONS Research Institute for Health Sciences and Technology, University of Mons (UMons), 7000 Mons, Belgium; jerome.lechien@umons.ac.be

**Keywords:** PANDAS, adenotonsillectomy, ENT, OCD, behavioral disorders, orobuccal disorders, infection

## Abstract

Background: Several treatment options have been proposed for pediatric acute-onset neuropsychiatric syndrome/pediatric autoimmune neuropsychiatric disorder associated with streptococcal infection (PANS/PANDAS). Still, no clear therapeutic protocol has been recognized to prevent these neuropsychiatric diseases. The study aims to report on the literature evidence and different treatment strategies related to these disorders. Methods: We analyzed the last 20 years’ English language literature and performed a comprehensive review of the PANS/PANDAS treatment, including studies reporting OCD outcomes post-treatment follow-up. Results: We covered 11 articles in our systematic literature review for a total of 473 patients, of which four studies included 129 surgical subjects and seven papers with 326 medically treated patients. Pooled outcomes analysis, surgical and medical treatment reported an OCD reduction, but no statistical significance was obtained (*p* < 0.05 for both). Conclusions: Surgical therapy in selected patients can lead to promising results, although further evidence is needed. On the other hand, the role of medical therapy remains controversial, often due to the lack of univocal curative protocols and variable responses depending on the drug used and the timing of administration. Therefore, further investigations are necessary to clarify the most appropriate therapeutic procedure.

## 1. Introduction

Post-infectious autoimmune and neuro-inflammatory events have been recognized to cause acute childhood neuropsychiatric disorders and have been designed with different terms, but all presented with similar clinical-pathogenetic manifestations. These disorders were previously indicated as paediatric infection-triggered autoimmune neuropsychiatric disorder (PITND); pediatric autoimmune neuropsychiatric disorder associated with streptococcal infection (PANDAS); paediatric acute-onset neuropsychiatric syndrome (PANS); and childhood acute neuropsychiatric syndrome (CANS) [1,2,3,4,5].

The clinical recommendations of the PANS Consensus Conference clarified the clinical evaluation and diagnostic criteria of patients with young pediatric acute-onset neuropsychiatric syndrome (PANS) [3].

PANS describes a clinical situation defined by the sudden and dramatic onset of obsessive-compulsive disorders or severely restricted food intake, associated with acute onset neuropsychiatric symptoms such as anxiety, emotional lability and/or depression, irritability, aggression, and/or strongly oppositional behavior behavioral regression (development), deterioration in school performance or memory impairment. On the other hand, the PANDAS subgroup is defined by an acute prepubertal onset of tics or OCD symptoms associated with GAS infection and specific neuropsychiatric symptoms. It is distinguished from PANS by a sudden onset, and episodic course and tics have an “off/on” and increasing/decreasing course. While PANDAS has a specific infectious pathogen responsible, PANS foresee different microbes possibly implicated in the genesis of the disorders postinfectious neurological such as H1N1 influenza, Epstein Barr virus, and *Borrelia burgdorferi* (Lyme) disease [4,6,7,8].

The infectious etiology in OCD has been suspected based the various pieces of evidence reported in the literature, involving viral or bacterial agents at the basis of the etiopathogenesis. These disorders linking OCD and infections have been described by Pavone et al., presenting two adolescents who acutely developed new OCD, neuropsychiatric, and motor dysfunction symptoms consistent with PANS 2 weeks after a diagnosis of COVID-19 [4]. The term PANS/PANDAS has been connected with a clinical condition in children and adolescents presenting with a sudden onset of various neuropsychiatric disorders, including obsessive-compulsive disorder (OCD), severely restricted food intake, anxiety, and inattention deficit hyperactivity disorder (ADHD). Therefore, diagnostic criteria have been proposed in order to allow a clear identification of individuals affected by PANS/PANDAS and consist of the onset of childhood/adolescent-related obsessive-compulsive disorder or severe restrictive eating, associated with at least two of the following neuropsychiatric disorders such as anxiety, emotional lability, and depression, irritability, aggression or strongly oppositional behavior, behavioral and developmental regression [5]. Other disturbances may include a deterioration in school performance, sensory or motor difficulties, somatic signs or symptoms, including sleep disturbance, enuresis, or increased urinary frequency. These disturbances should not be better explained or be related to a known neurologic or medical disorder [4,5,6,7,8,9,10].

The simultaneous presence of additional neuropsychiatric symptoms has been associated with similarly severe and acute onset, such as anxiety, emotional lability or depression, irritability, aggression, and severely oppositional behaviors, and sensory or motor difficulties [11,12]. In addition, somatic signs or symptoms may also be reported, including sleep disturbances, enuresis, or urinary frequency [12].

At the diagnosis of PANS/PANDAS, several types of treatment have been proposed and used according to the prevalent clinical signs and the severity of the disturbances. Antibiotics (penicillin V, azithromycin), anti-inflammatory drugs (cyclooxygenase (COX) inhibitors, corticosteroids), immunomodulating treatments (intravenous immunoglobulin –IVIG. plasma exchange) are the most applied treatment singularly or in association [13,14,15,16,17,18,19,20,21,22,23,24,25,26,27].

To clarify the role of the various types of treatments on PANS/PANDAS, we performed the meta-analysis to reveal how some therapeutic interventions may influence the course of OCD, one of the main features of these disorders.

## 2. Materials and Methods

We used the PRISMA statement to conduct the systematic review and meta-analysis [28], while the PICOTS statements for the method presentation [29].

In particular, the following criteria were considered: Participants (PANS/PANDAS children/adolescents); Intervention (Adenotonsillectomy); Control (Medical treatment); Outcome (obsessive-compulsive disorder improvement), and study type (observational study). In addition, language, publication date, and publication status were imposed as research restrictions.

The primary outcome was a significant improvement in the reduction at the clinical examination of OCD at the post-treatment follow-up due to the accuracy and clearness in comparing the results on behavioral disorders related to the syndrome. Moreover, additional parameters reported in the studies were recognized as secondary outcomes. We included all the studies that met the consequent criteria: (1) Original articles; (2) Articles published in the English language; (3) Studies including PANS/PANDAS individuals undergoing total surgical or medical treatment; (4) Studies that reported detailed information on post-treatment OCD outcomes, several therapeutical modalities, and patient’s comorbidities; (5) Excluded from the study were editorials, letters to the editor, case reports, erratum, duplicates or reviews.

### 2.1. Protocol Data Extraction and Outcomes

The authors A.M., F.N., and D.C. investigated the literature data, resolving any disagreements through discussion. We thus analyzed the included data to achieve all the available clinical elements and guarantee eligibility among subjects enrolled. The main patient’s features, including symptoms, age, gender, validated questionnaires, treatment modalities (surgical or medical), were collected. The following information was also reported: study design, author data, year, sample size, statistical analysis, outcomes, and conclusions. We contacted the included ‘authors if the required data were not complete utilizing the corresponding author’s email or Research Gate (http://www.researchgate.net/) (accessed on 2 November 2021).

#### 2.1.1. Electronic Database Search

Three different authors examined the PubMed, Scopus, and Web of Science electronic databases for studies on OCD outcomes in PANS/PANDAS patients undergoing different treatment modalities in the last 20 years’ literature (from 1 December 2001, to 1 June 2021). We used MeSH, Entry Terms, and related keywords. The following search keywords were adopted: “PANS and PANDAS patients’’, ‘‘OCD’’, ‘‘adenotonsillectomy’’, ‘‘tonsillectomy’’, ‘‘obsessive-compulsive disorders’’, ‘‘surgical treatment”, ‘‘medical treatment’’ separated by Boolean operator ‘‘and/or’’.

The “Related articles” option was also performed on the PubMed homepage. We used reference manager software (EndNote version X7^®^, Thomson Reuters: Philadelphia, PA, USA 2015) to collect references and remove duplicates. The investigators examined titles and abstracts of papers available in the English language. The full texts identified were screened for original data, and the references were retrieved to check manually other relevant studies.

#### 2.1.2. Statistical Analysis

According to the approved reporting items’ quality requirements for systematic review and meta-analysis protocols (PRISMA) declaration, a systematic review was conducted [30]. The studies’ quality assessment (QUADAS-2) instrument was employed to estimate the included studies’ and descriptively present the risk of bias [31]. The observational studies’ potential risk of bias was estimated using the Joanna Briggs Institute Critical Assessment Checklist for Observational Studies.

Statistical analysis was conducted through the statistical software (SPSS, IBM Statistics for Windows, version 25.0, IBM Corp., Armonk, NY, USA 2017). Random-effects modeling (standard error estimate = inverse of the sample size) was used to assess summary effect measures by 95% confidence intervals (CI). Consequently, forest plots were generated through the Review Manager software (REVMAN) version 5.4 (The Nordic Cochrane Centre: The Cochrane Collaboration, Copenhagen, Denmark). The inconsistency (I2 statistic) was thus calculated and the values for low inconsistency = 25%, moderate inconsistency = 50%, and high inconsistency = 75% were established [31].

## 3. Results

### 3.1. Retrieving Researches

We identified 418 potentially relevant studies through the systematic review of the literature (Figure 1).

After removing the duplicates and applying the criteria listed above, an overall number of 415 records screened were potentially relevant to the topic. We excluded all the studies not matching inclusion criteria through the records analysis and the following articles’ full-text screening (*n* = 404). Thus, the remaining 11 papers were included in qualitative synthesis papers for the data extraction. After the meta-analysis established criteria, we excluded five papers (partial data) and considered six studies for quantitative analysis. The possible risk of bias is summarized as a graphical QUADAS-2 outcome in Figure 2.

We confirmed eligibility among the symptoms reported in the papers, including only OCD as a comparable parameter after the administered treatment.

The probable risk of bias in observational studies was assessed using the Joanna Briggs Institute Critical Assessment Checklist for Observational Studies (Figure 3) [30].

### 3.2. Patients Features

We provided 11 articles in our systematic literature review for a total of 473 patients surgically or medically treated. The patients’ average age was 9.18 ± 1.47 years [14,15,20,21,22,23,24,32]. In addition, a significant difference in sex ratio was reported (59.9% male vs. 40.1% female).

## 4. Surgical Treatment

Four papers included 129 patients surgically treated, reporting both pre-and post-treatment OCD ratio (event/total) according to fixed effect model (Table 1) [13,22,23,32].

All patients underwent adenotonsillectomy for PANDAS, and the results obtained were compared with a control group. On the pooled analysis, greater improvements in OCD occurred in patients undergoing surgery than in controls (80/129; 62% vs. 91/160; 56.87%); however, no statistical significance was reached (*p* < 0.65) (Figure 4).

The analysis using fixed-effects modeling for 150 surgical procedures (3 papers) [13,22,32] demonstrated an OR of 2.17 [95% CI 1.11, 4.27], an overall effect Z score = 2.25 (*p* = 0.02), Q statistic *p* = 0.39 and not statistically significant heterogeneity I2 = 0% as described in Figure 5.

## 5. Medical Treatment

OCD outcomes after medical treatment were reported in seven papers (326 subjects), of which four papers used antibiotics [15,16,20,21], two papers intravenous immunoglobulin (IVIG) [19,20], one paper a non-steroidal anti-inflammatory (NSAID) [14], and one paper (54 patients) corticosteroids [18].

At pooled analysis, OCD improvement was reported in 150/263 patients treated. Moreover, the analysis using fixed-effects modeling for the OCD outcomes found an OR of 1.86 [95% CI 0.81, 4.28]. The reported overall effect Z score was 1.47 (*p* < 0.14), Q statistic *p* = 0.41 (no significant heterogeneity), I2 = 0% as described in Figure 5.

## 6. Discussion

The PANDAS concept was first stated by Sweedo et al. in 1998 [1]. Among the various diagnostic hypotheses reported in the literature, streptococcal and other infections have been shown to trigger the development of symptoms such as OCD, tics and behavioral disturbances [2,3,5]. Therefore, numerous studies have been performed on neurological outcomes in PANDAS patients undergoing medical or surgical treatments, with the principle of adenotonsillar infection as the primary target [13,14,15,16,17,18,19,20,21,22,23,24,25,26,27,32,33]. Pavone et al. examined whether adenotonsillectomy could impact both disease remission and affect the clinical course, streptococcal antibody titers, neuronal antibodies, or the clinical severity of the obsessive-compulsive disorder (OCD) [13]. Surgery did not affect symptom progression, streptococcal and neuronal antibodies or clinical severity of neuropsychiatric symptoms with comparable results for remission (17 surgical vs. 14 non-surgical; *p* = 0.29) and disease recurrence (39 surgical vs. 50 controls; *p* = 0.09) [14]. Conversely, promising results have been reported later by Demesh et al. [22]. The authors noted that nine surgically treated patients achieved clinically significant relief, including those who had no response to antibiotic therapy alone (*p* = 0.03). However, the sample of enrolled patients constitutes a limitation for the reported study, reducing the significance of the evidence demonstrated.

However, the evidence that analyzes the efficacy of surgical treatment remains scarce, especially in the differentiation of outcomes based on the type of intervention administered (tonsillectomy, adenoidectomy or adenotonsillectomy). Furthermore, the results of the surgical treatment should be compared with an adequate control group composed of a homogeneous sample treated with a medical therapy validated in the literature.

Prasad et al. attempted to compare the outcomes achieved through surgical treatment alone versus those obtained from the combined surgical approach with intravenous immunoglobulins [23]. However, the authors enrolled a small group of patients, dividing them into three different treatment arms: tonsillectomy and adenoidectomy (AT) (*n* = 28), AT plus intravenous immunoglobulin (IVIG) (*n* = 22), or non-surgical treatment (*n* = 10). Although caregivers did not report a decrease in symptom frequency depending on the type of treatment except choreiform movement (*p* = 0.0296), TA was shown to be the treatment with the greatest symptom impact for patients (*p* = 0.05).

Another limitation of the study was that it did not administer only the treatments described in the three study arms. These patients benefited from further treatments such as antibiotics (*n* = 60, 100%), rituximab (15%), steroids (20%), and plasma exchange (10%) which constituted a potential risk of bias.

Another issue of concern remains the effectiveness of surgical treatment on disease prevention by influencing the onset of neuropsychiatric symptoms. Nevertheless, when examining the literature, the only quantifiable outcome remains the change in OCD before and after treatment as reported in our systematic review, not allowing a quantitative analysis of the results and, therefore, not drawing valid conclusions.

In this regard, Murphy et al. reported in 2013 poor prevention of neuropsychiatric symptoms (e.g., OCD or tics) based on patient surgical status (*p* = 0.71), inferring a mismatch between tonsillectomy and course of OCD/tics or concentration of streptococcal antibodies [32]. In our meta-analysis, we found interesting data on OCD remission. Although no statistical significance was achieved (*p* < 0.65), surgery demonstrated better OCD outcomes than control (80/129, 62% vs. 91/160, (56.87%).

We interpreted these data as interesting because although statistical significance has not been achieved, this may be simply due to the lack of evidence in the literature on this subject, particularly insufficient sample sizes, study protocols, and non-standardized selection criteria that do not allow adequate comparisons.

Using the same etiopathogenetic principle, antibiotic therapy has been proposed as a treatment for PANDAS patients, boasting a dual purpose, acting both on the prevention of the disease and relapses [15,16,20,21]. Murphy et al. in 2002 reported promising medical therapy results, with a rapid resolution of the symptoms of OCD, anxiety and tics that occur after 14 days of appropriate antibiotic treatment in 6/12 patients with PANDAS [21]. The authors put forward an interesting point of view regarding the management of related symptoms. GABHS serological tests have been used as an objective evaluation of response to treatment and the eradication of the germ by antibiotics has shown efficacy in the resolution of OCD symptoms as well as any relapse after acute streptococcal infection. Therefore, the authors consistently with what was hypothesized in the study, they obtained the predetermined outcomes, even though they enrolled an insufficient sample of patients.

In reverse, different studies instead focus on the limits of the efficacy of antibiotics in prophylaxis, affirming instead the therapeutic benefits during acute episodes. Snider et al. reported a significantly reduced number of exacerbations of neuropsychiatric symptoms in both penicillin and azithromycin-treated patients (*p* < 0.01 for both) [15]. Subsequently, the possible role of antibiotic therapy was hypothesized by a study comparing the severity of OCD on the Clinical Global Impression Scale (CGI-S). However, it should be noted that the authors in this case did not correctly correlate the number of exacerbation and the eradication of the germ to the reduction of specific symptoms.

In 2017 Murphy et al. comparing two different medical treatment for PANDAS patients, demonstrated significant reductions in the azithromycin group (*n* = 17) matched to the placebo group (*n* = 14) (*p* = 0.003) [16].

Other authors have instead evaluated the effects of reducing oropharyngeal and nasosinus inflammation after treatment with non-steroidal anti-inflammatory drugs in an attempt to improve neuropsychiatric symptoms, reporting unpromising results [18,19]. The lack of efficacy has probably been interpreted in the role of the eradication of the germ in the resolution of the pathology, regardless of the reduction of the episodes of inflammation. In particular, Brown et al. in 2017 found no significant improvement in OCD in patients with PANDAS (*n* = 54) compared to placebo (*n* = 44) (*p* = 0.99) [18].

In response to a Group A Streptococcal infection, cross-reactive antibodies have been hypothesized in the etiology and pathogenesis of PANDAS in susceptible individuals by reacting to cell wall components and neuronal proteins of the basal ganglia [34,35,36,37,38]. For this reason, some authors have tested the possible role of intravenous immunoglobulins (IVIG) on the clinical course of the disease [19,24].

Although Williams et al. demonstrated that IVIG was safe and well-tolerated, differences between groups in the double-blind comparison did not demonstrate the superiority of IVIG over placebo [19,24]. In this regard, the reported outcomes certainly suffered from the disadvantage of small sample sizes and the lack of specific biomarkers predicting a positive response to immunotherapy.

Exploiting the same concept of higher concentrations of cross-reactive antibodies in acute serum samples, therapeutic plasmapheresis (TPE) has been proposed as an alternative treatment. However, the support for the plasmapheresis use remains limited to a few reports and controlled-placebo trials with a lessened population, demonstrating symptom improvements mainly in the short-term follow-up.

Latimer et al. proposed therapeutic plasmapheresis (TPE) to treat 35 severely ill children and adolescents with PANDAS disorders [38]. The authors reported an average improvement of 65% at six months post-TPE and 78% at long-term follow-up. However, the sample enrolled by the authors was not homogeneous, with possible confounding variables not adequately identified. In agreement with what has been stated, our meta-analysis performed showed an odd ratio clearly in favor of the experimental group of 0.98 (025, 3.89) for the fixed effect of the medical subgroup for Brown et al. and 1.91 (0.43, 8.48) for Williams et al. However, the overall effect test on sub analysis did not reach statistical significance (*p* = 0.14), probably due to the need to include large shorts from PANDAS patients in the analysis.

The therapeutic effect of corticosteroids, also proposed in the treatment of PANDAS, seems to be affected by the timing of drug administration. Indeed, while early administration appears to be associated with shorter flare-up periods (*p* < 0.001) [39], longer administrations are associated with better control of neuropsychiatric symptoms (*p* = 0.014) [18]. The probable therapeutic hypothesis is the stabilization of the chronic inflammatory state induced by the streptococcal infection and the maintenance of a low antibody titre, although in the literature there are insufficient data in this regard.

At the overall pooled analysis performed for the subgroup of medical patients, the improvement in OCD was recorded in 150/263 (57.03%) patients. However, at the subsequent meta-analysis of the data by subgroup fixed effects modeling although an OR of 1.86 was found [95% CI 0.81, 4.28], the overall effect Z score 1.47 (*p* < 0.14) heterogeneity I2 = 0% (*p* = 0.41) were not significant. It should be noted that several medical approaches have been adopted without any standardized protocol, which represents a major limitation for the pooled analysis of the studies included.

From our analysis of the evidence reported to date on the treatment of patients with PANDAS, it is clear how evident limitations are present in the literature. The scientific evidence reported is scant, not sufficient to propose a clear line of treatment for PANS/PANDAS and related disorders [9,33,40,41,42,43].

### Study Limitations

The available studies are characterized in almost all cases by study cohorts that are too small, with low-evidence study designs such as case reports or uncontrolled retrospective studies. Furthermore, if there are prospective controlled or randomized studies, these do not present clear patient selection criteria or standardized treatment protocols. Therefore, the treatments administered are too heterogeneous and the results obtained are arbitrarily evaluated both in prevention and after treatment, without a univocal rationale. In addition, although each author proposes a different line of treatment, most do not propose a specific biomarker that evaluates the effectiveness of the therapy administered but rather uses those previously known.

Therefore, the main limitations set out so far require the implementation of new research protocols, designed with strict criteria for selecting patients and large study samples, to eliminate any confounding factor in the outcome of the treatment and to standardize and standardize the treatments administered according to appropriate analysis of the evidence present to date.

## 7. Conclusions

Although PANS/PANDAS are disorders with well-defined diagnostic criteria, the proposed therapeutic protocols report conflicting data to date. Furthermore, the effectiveness of medical and surgical approaches on movements and behavioral disorders often does not reach significant differences and is frequently affected by the timing of administration. Finally, it is necessary to consider that the evidence in the literature does not follow a unified therapeutic protocol, leading to poor enrolled samples and unsatisfactory results.

## Figures and Tables

**Figure 1 children-09-00155-f001:**
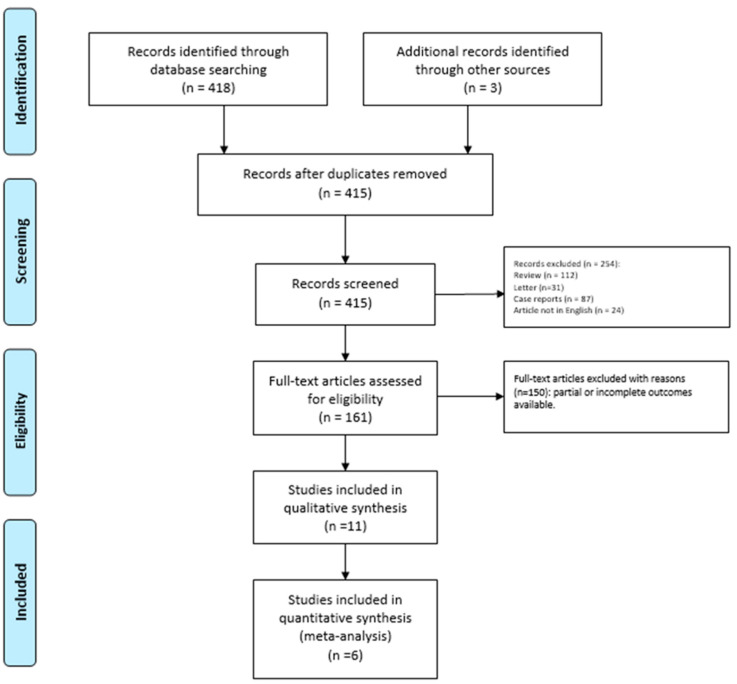
PRISMA Flow-diagram.

**Figure 2 children-09-00155-f002:**
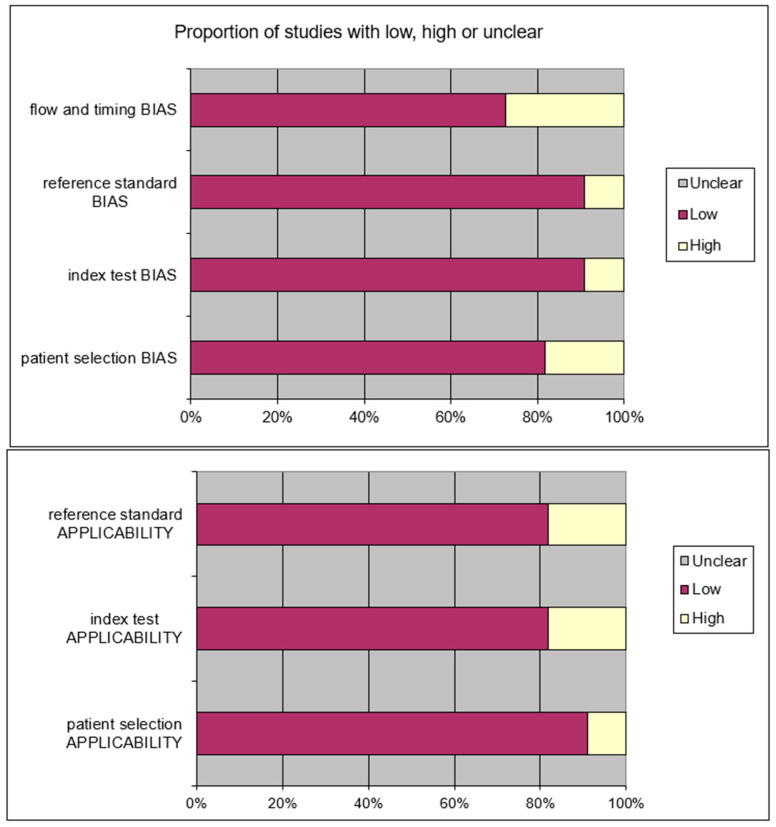
QUADAS2 risk of bias.

**Figure 3 children-09-00155-f003:**
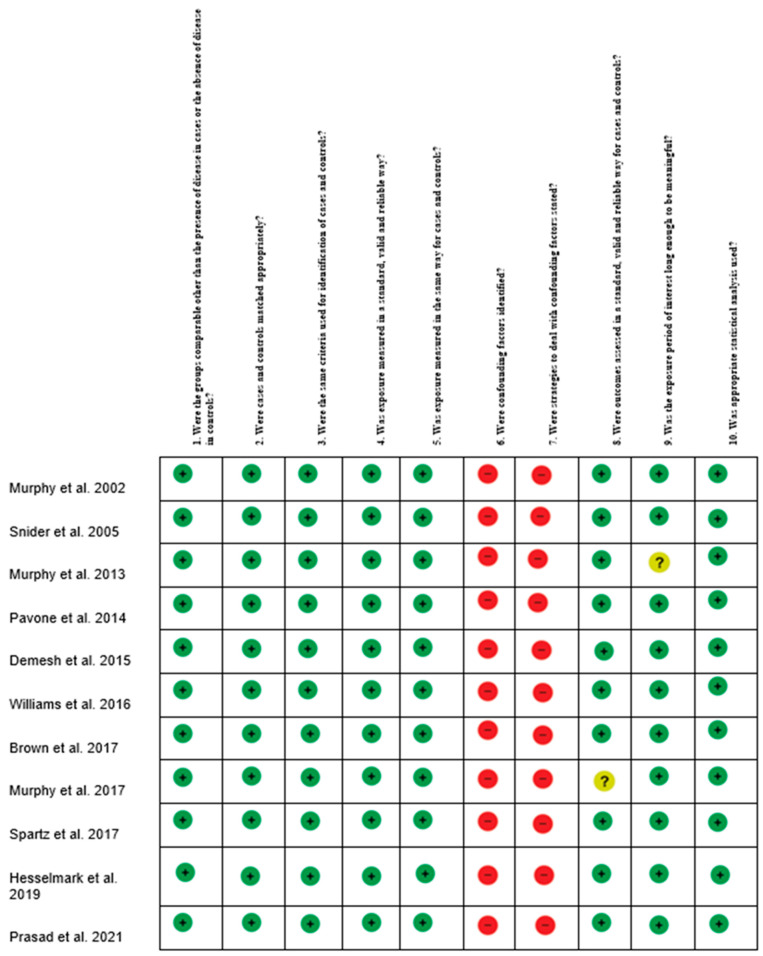
Risk of Bias summary author’s judgments for each included study, assessed by the Joanna Briggs Institute (JBI). Critical Appraisal Checklist for Case-Control studies.

**Figure 4 children-09-00155-f004:**
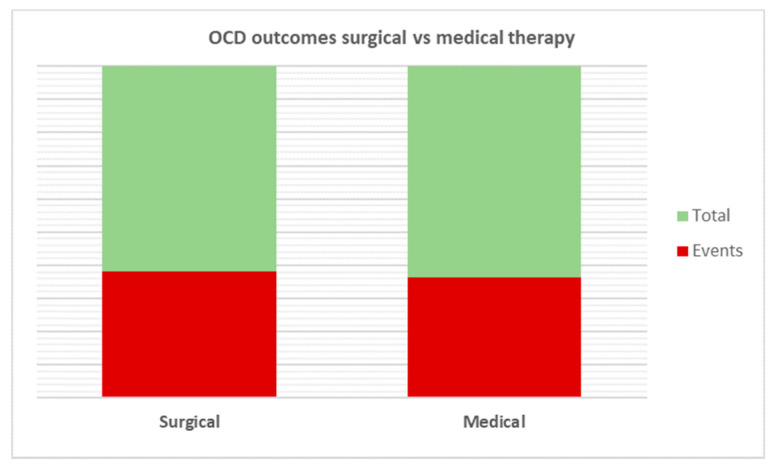
Comparative analysis of OCD outcomes after treatment. No statistical significance was reached among the two therapies analyzed (*p* = 0.65).

**Figure 5 children-09-00155-f005:**
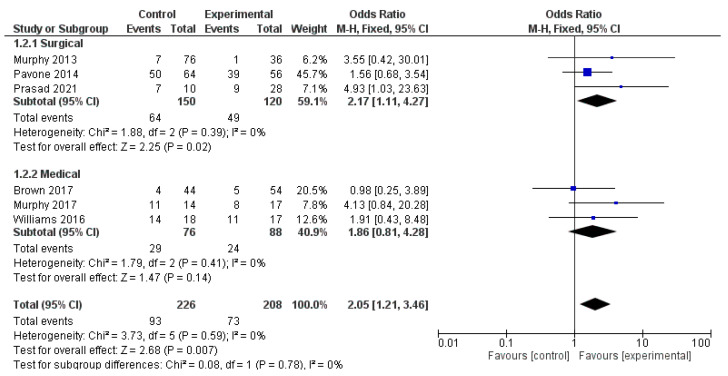
Forest-plot analysis.

**Table 1 children-09-00155-t001:** Main studies’ Features. Abbreviations: PANDAS: Pediatric Autoimmune Neuropsychiatric Disorder associated with streptococcal infections; IVIG, intravenous immunoglobulin; M, male; F, Female; OCD, Obsessive-compulsive disorder; NSAID, Non-steroidal anti-inflammatory.

Surgical							
Authors, Year, Reference	Study Design	Patients	Age (Mean ± SD/Range)	Gender	Treatment	OCD Treatment Group	OCD Control Group
Murphy et al., 2013 [32]	Prospective controlled study	43 PANDAS vs. 69 Healthy	9.18 ± 2.38	68M vs. 44F	36 Surgery vs. 76 N-Surgery	35/36 (OCD)	69/76 (OCD)
Pavone et al., 2014 [13]	Prospective study controlled	120 PANDAS	11.05 ± 1.2	63M vs. 57F	56 Surgery vs. 64 N-Surgery	17/56 (OCD)	14/64 (OCD)
Prasad et al., 2021 [20]	Retrospective study controlled	60 PANDAS	-	-	28 Surgery vs. 10 N-Surgery	19/28 (OCD)	3/10 (OCD)
Demesh et al., 2015 [22]	Retrospective study controlled	10 PANDAS	6.5	8 M vs. 2 F	9 Surgery & N-Surgery (Antibiotics) vs. 1 N-Surgery	9/9 (OCD)	5/10 (OCD)
**NonSurgical**							
Murphy et al., 2017 [16]	Prospective randomized controlled study	31	8.26 ± 2.78	20M vs. 11 F	17 Azithromycin vs. 14 Placebo	9/17 (OCD)	3/14 (OCD)
Snider et al., 2005 [15]	Prospective randomized controlled study	23	7.9 ± 1.3	15 M vs. 8F	11 Penicillin vs. 12 Azithromycin	6/11 (OCD)	11/12 (OCD)
Spartz et al., 2017 [17]	Retrospective study	77	8.3 ± 3.6	42 M vs. 35 F	77 NSAID	32/77 (OCD)	-
Hesselmark et al., 2019 [20]	Retrospective controlled study	53	7.9 (1–20)	33 M vs. 20 F	46 Antibiotics vs. 17 IVIG	19/46 (OCD)	12/17 (OCD)
Brown et al., 2017 [14]	Retrospective controlled study	95	7.8 ± 3.8 Treatment vs. 8.6 ± 3.2 Placebo	-	54 Corticosteroids vs. 44 Placebo	49/54 (OCD)	40/44 (OCD)
Murphy et al., 2002 [21]	Prospective controlled study	12	7 (range 5.4–10.11)	7 M vs. 5 F	5 Penicillin vs. 1 Amoxicillina/Clavulanic vs. 6 Cephalosporin	6/12 (OCD)	-
Williams et al., 2016 [19]	Prospective randomized controlled study	35	placebo 9.61 ± 2.32 IVIG 8.99 ± 2.37	23 M vs. 12 F	17 IVIG vs. 18 Placebo	6/17 (OCD)	4/18 (OCD)

## Data Availability

Not applicable.

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
