# Peer review of "Obsessive-Compulsive Disorder in PANS/PANDAS in Children: In Search of a Qualified Treatment—A Systematic Review and Metanalysis"

_children, 2022, doi:10.3390/children9020155_

Round 1

Reviewer 1 Report

Authors respond to all my comments. I have no further remarks.

Reviewer 2 Report

Nice study. I like it a lot since it is so helpful to clinicians in this field. Research design is appropriate. Data is solid. Good introduction and discussion.

This manuscript is a resubmission of an earlier submission. The following is a list of the peer review reports and author responses from that submission.

Round 1

Reviewer 1 Report

Authors analyzed the last 20 years' English language literature and performed a comprehensive review of the PANS/PANDAS treatment, including studies reporting OCD outcomes at post-treatment follow-up. The study is interesting and fits to the journal content. However, some methodological aspects are for me unclear, especially reason for evaluating only OCD symptoms (see detailed comments below). Therefore, I suggest major revision of the manuscript

-Authors concentrate on the OCD during the PANS/PANDAS so I suggest changing the title to highlight the analysis of these element in PANS/PANDAS (for example: OCD in the course of PANS/PANDAS in children: in search of a qualified treatment. A systematic review).

-why did authors not include other psychiatric symptoms like ADHD in their analyses?

-I suggest to authors to write a little bit more in the introduction section about other symptoms of PANS/PANDAS like tics, ADHD. Are they specific (Tipton P. Neurol Neurochir Pol 2019; 53 (5): 315–316)?

-There are several typos like in page 9 line 199 “significantdifferences”

Author Response

Reviewer 1

Authors analyzed the last 20 years' English language literature and performed a comprehensive review of the PANS/PANDAS treatment, including studies reporting OCD outcomes at post-treatment follow-up. The study is interesting and fits to the journal content. However, some methodological aspects are for me unclear, especially reason for evaluating only OCD symptoms (see detailed comments below). Therefore, I suggest major revision of the manuscript:

-Authors concentrate on the OCD during the PANS/PANDAS so I suggest changing the title to highlight the analysis of these element in PANS/PANDAS (for example: OCD in the course of PANS/PANDAS in children: in search of a qualified treatment. A systematic review).

Response: dear revisor, we share your opinion regarding the OCD parameter analyzed. We are fully aware of the complexity of PANDAS patient disorders. However, after thorough research of the various outcomes, the only possible parameter to retrieve was adequately reported and comparable with the various studies included in the analysis. OCD disorders are, therefore, the only easily similar parameter of these patients. As suggested, we have thus changed the title of the manuscript to ‘’Obsessive-compulsive disorder in PANS / PANDAS in children: in search of a qualified treatment. A systematic review.’’

-why did authors not include other psychiatric symptoms like ADHD in their analyses?

Response: Dear Reviewer, As described in the previous answer, the only parameter included in the analysis was the OCD. In order to justify what has been stated in the paper, we have included a sentence in the methods and results respectively that justifies the selection of the OCD parameter among those available:

‘’due to the accuracy and clearness in comparing the results on behavioral disorders related to the syndrome

We confirmed eligibility among the symptoms reported in the papers including only OCD as a comparable parameter after the administered treatment’’.

-I suggest to authors to write a little bit more in the introduction section about other symptoms of PANS/PANDAS like tics, ADHD. Are they specific (Tipton P. Neurol Neurochir Pol 2019; 53 (5): 315–316)?

Response: as suggested we added a description of other symptoms ans signs associated to PANDAS

Disorders. In addition we cited two different papers related: Tipton P. Neurol Neurochir Pol 2019; 53 (5): 315–316 and Wilbur C, Bitnun A, Kronenberg S, et al. PANDAS/PANS in childhood: Controversies and evidence. Paediatr Child Health. 2019;24(2):85-91. doi:10.1093/pch/pxy145

-There are several typos like in page 9 line 199 “significantdifferences”:

response: dear revisor, as suggested all the text was corrected from typos and grammatical editing was performed.

Reviewer 2 Report

Dear Editor, dear Authors,

First of all, thank you for letting me review this work.

This work was about the treatment of the PANS entity, an entity that have been highlighted as a possible etiology of OCD. The authors led a review and a meta-analysis to look for the efficiency of the different treatment tested in this disorder.

Strengths of the work :

  • The subject of this article is really interesting and such a work is needed in the controversial PANS field.
  • The methodology is good and the work seems really complete.

Minor concerns:

  • Line 41: I understand you want to cite yourself, but I am not sure that it is relevant here. You mentioned the different terms and entities linked to PANS, then you described what is PANS, but between this two parts you mentioned a case report without any logical link. At least, you should say something like “…infectious etiology in OCD has suspected through different observations like this case report or this case report with different virus or bacteria involved. These disorders linking OCD and infections has been indicated as…”. In this way you can cite yourself (adding others case report, only one case report has no sense), it would seem less weird.
  • Line 48: I’m not sure of your PANS definition. Indeed, it seems that OCD is not systematically linked with restrictive eating in PANS as you mean, but that it is either OCD OR restrictive eating AND others symptoms (if I’m wrong, please develop).
  • Line 71: Did you mean you have limited your research to a specific period?
  • For the PRISMA figure: I guess you made a screen shot but be careful, beside the top legend, there is a bar: “PRISME Flow Diagram|”. Same remark for the figure 3 where unknown words are represented with red line.
  • Line 149: What is the ratio you are talking about, what on what? Did you mean a YBOCS severity ratio or something like that?
  • Line 170: What you mean is really clear but I think the sentence is really bad. You should rewrite this sentence.
  • Line 196: Tics should not be in capital, it is not an abbreviation, it is just a movement disorder.
  • Line 208: “Although no statistical significance was reached (p < 0.65), surgery demonstrated better OCD outcomes than control 209 (80/129, 62% vs. 91/160, (56.87%).” If it is not significative, you can not say that. I understand that in a absolute way, there is a difference but this difference could be due to the chance.
  • Line 223: I guess there is a mistake in this sentence. The same remark is true in line 227. In both case, the p values seem contradictory to what you explain.

Major concerns:

  • I have to recognize that I am very disappointing with your work. I do not criticize your results or conclusion of course but the discussion is clearly not a discussion.
    • You just mentioned in a row a list of studies without making any separation between the different part. The most representative fact of this limit is that your discussion is contained in only… one paragraph!! Like a big square! Which makes it clearly difficult to read.
    • Furthermore, there is no personal interpretation of these studies. I mean you listed the results but we do not know what were the studied populations, what is your interpretation concerning the discrepancies between the different results etc… It is clearly not a discussion. It is a list. I think your discussion should be in the result part and in a clearer way (with different paragraph according to the subject: surgical treatment or medical treatment, and medical treatment divided according to the type of medical treatment used) and, then, that you should discuss them in the discussion.
  • I understood most of the articles you included did not allow you to be meta analyzed as they should not meet your inclusion criteria but, each time, you should explain why because you cited a lot of articles on medical treatment and only 3 are in your quantitative part. This imbalance should be discussed (in the discussion part for example, for example highlighting the global poor quality of the studies on the subject that limited your ability to meta analyze them, it is only an example).

As a conclusion: your work is an interesting one, and the methodology of quality but the writing of your work really really needs to be improved. So please, write your article again with clear different parts and, please please please, take the time to discuss your results (why the discrepancies, what was bad and good in the different studies. For the surgical treatment I have the feeling that you mixed results of surgery as prevention and as treatment, maybe I m wrong, in this case your line 206/207 is unclear, etc etc…). It seems that you wanted to get rid of this work, but I would like you to be published, so write it again and it will be a good paper!

Thank you again, sorry to be a bit harsh, but this interesting work deserves to be better written. Sorry for my terrible English!

Author Response

Reviewer 2

This work was about the treatment of the PANS entity, an entity that have been highlighted as a possible etiology of OCD. The authors led a review and a meta-analysis to look for the efficiency of the different treatment tested in this disorder.

Strengths of the work :

  • The subject of this article is really interesting and such a work is needed in the controversial PANS field.
  • The methodology is good and the work seems really complete.

Response to the revisor: thanks for the opinions. We’re pleased to receive a good opinion for our efforts.

Minor concerns:

  • Line 41: I understand you want to cite yourself, but I am not sure that it is relevant here. You mentioned the different terms and entities linked to PANS, then you described what is PANS, but between this two parts you mentioned a case report without any logical link. At least, you should say something like “…infectious etiology in OCD has suspected through different observations like this case report or this case report with different virus or bacteria involved. These disorders linking OCD and infections has been indicated as…”. In this way you can cite yourself (adding others case report, only one case report has no sense), it would seem less weird.

Response: dear revisor, we share your suggestions and thus we added the text in the introduction: The infectious etiology in OCD has been suspected among the various evidence reported in the literature, involving viral or bacterial agents at the basis of the etiopathogenesis. These disorders linking OCD and infections have been described by Pavone et al., presenting

  • Line 48: I’m not sure of your PANS definition. Indeed, it seems that OCD is not systematically linked with restrictive eating in PANS as you mean, but that it is either OCD OR restrictive eating AND others symptoms (if I’m wrong, please develop).

Dear revisor, according to your suggestion we clarified the definition of PANS according to the latest guidelines that were added in the references as below: The term PANS/PANDAS has been connected with a clinical condition in children and adolescents presenting with a sudden onset of various neuropsychiatric disorders, including obsessive-compulsive disorder (OCD), severely restricted food intake, anxiety, and inattention deficit hyperactivity disorder (ADHD). Therefore, diagnostic criteria have been proposed in order to allow a clear identification of individuals affected by PANS/PANDAS and consist of the onset of childhood/adolescent-related obsessive-compulsive disorder or severe restrictive eating, associated with at least two of the following neuropsychiatric disorders such as anxiety, emotional lability, and depression, irritability, aggression or strongly oppositional behavior, behavioral and developmental regression.

Thienemann M, Murphy T, Leckman J, et al. Clinical Management of Pediatric Acute-Onset Neuropsychiatric Syndrome: Part I-Psychiatric and Behavioral Interventions. J Child Adolesc Psychopharmacol. 2017;27(7):566-573. doi:10.1089/cap.2016.0145

  • Line 71: Did you mean you have limited your research to a specific period?

Response: dear revisor, as reported in the methods and highlighted, we limited the research to the following period ‘’ modalities of the last 20 years' literature (from Dec 1, 2001, to Jun 1, 2021)’’

  • For the PRISMA figure: I guess you made a screen shot but be careful, beside the top legend, there is a bar: “PRISME Flow Diagram|”. Same remark for the figure 3 where unknown words are represented with red line.

Response: dear revisor, thanks for the suggestions, we modified the two figures removing the errors and adding captation.

Line 149: What is the ratio you are talking about, what on what? Did you mean a YBOCS severity ratio or something like that?

Response: dear reviewer, it would certainly have been interesting to exploit a useful index such as the Yale-Brown Obsessive-Compulsive Scale or the Yale Global Tic Severity Scale (YGTSS). However, there was not enough data to make comparisons using these scales. Therefore, the ratio used was according to the fixed effect consisting of the number of events (presence of patients with OCD in the sample) on the total number of patients before and after the treatment administered. Thus, we clarified in the results section the reason behind our decision: ‘’OCD ratio(event/total) according to fixed effect model’’

  • Line 170: What you mean is really clear but I think the sentence is really bad. You should rewrite this sentence.

Response: thanks for the suggestions, we modified all the sentence as below: ‘’All patients underwent adenotonsillectomy for PANDAS, and the results obtained were compared with a control group. On the pooled analysis, greater improvements in OCD occurred in patients undergoing surgery than in controls (80/129; 62% vs. 91/160; 56.87%); however, no statistical significance was reached (p <0.65)’’

  • Line 196: Tics should not be in capital, it is not an abbreviation, it is just a movement disorder.

Response: thanks for the suggestion, we modified the capital.

  • Line 208: “Although no statistical significance was reached (p < 0.65), surgery demonstrated better OCD outcomes than control 209 (80/129, 62% vs. 91/160, (56.87%).” If it is not significative, you can not say that. I understand that in a absolute way, there is a difference but this difference could be due to the chance.

Response: dear revisor, thanks for the opinion, we reformulated the sentence. We only reported what the data analysis achieved as objectively as possible. The sentence 170 in our version of the paper is the same of the line 208. Maybe there was a typo.

  • Line 223: I guess there is a mistake in this sentence. The same remark is true in line 227. In both case, the p values seem contradictory to what you explain.

Response: dear revisor, thanks for the suggestions, we corrected the mistakes as below. The values were correct but not correctly expressed in the text. We modified as below: ‘Brown et al. in 2017 did not find significant OCD improvement in patients with PANDAS (n = 54) compared to placebo (n = 44) (p = 0.99). [125]. Conversely, Williams et al. in a randomized, controlled study on the effects of intravenous immunoglobulin (IVIG) treatment for PANDAS patients demonstrated the superiority of IVIG over placebo (p <0.0001).

Major concerns:

  • I have to recognize that I am very disappointing with your work. I do not criticize your results or conclusion of course but the discussion is clearly not a discussion.
    • You just mentioned in a row a list of studies without making any separation between the different part. The most representative fact of this limit is that your discussion is contained in only… one paragraph!! Like a big square! Which makes it clearly difficult to read.
    • Furthermore, there is no personal interpretation of these studies. I mean you listed the results but we do not know what were the studied populations, what is your interpretation concerning the discrepancies between the different results etc… It is clearly not a discussion. It is a list. I think your discussion should be in the result part and in a clearer way (with different paragraph according to the subject: surgical treatment or medical treatment, and medical treatment divided according to the type of medical treatment used) and, then, that you should discuss them in the discussion.
  • I understood most of the articles you included did not allow you to be meta analyzed as they should not meet your inclusion criteria but, each time, you should explain why because you cited a lot of articles on medical treatment and only 3 are in your quantitative part. This imbalance should be discussed (in the discussion part for example, for example highlighting the global poor quality of the studies on the subject that limited your ability to meta analyze them, it is only an example).

As a conclusion: your work is an interesting one, and the methodology of quality but the writing of your work really really needs to be improved. So please, write your article again with clear different parts and, please please please, take the time to discuss your results (why the discrepancies, what was bad and good in the different studies. For the surgical treatment I have the feeling that you mixed results of surgery as prevention and as treatment, maybe I m wrong, in this case your line 206/207 is unclear, etc etc…). It seems that you wanted to get rid of this work, but I would like you to be published, so write it again and it will be a good paper!

Response: dear editor, I’ve totally rewrote the discussion according to your suggestions, describing for each subgroups the specific evidences reported, our metanalysis outcomes, strenght and main limitations. At the end of the discussion we also added a paragraf on study limitations. Hope our job will be correct and clear. Thanks for your efforts and the work done.

Round 2

Reviewer 1 Report

Authors respond to all my comments. I have no further remarks. I suggest acceptance of the article.

Author Response

Reviewer 1

Dear Editor, thanks for the positive response. Our efforts were appreciated. Best regards.

Reviewer 2 Report

Dear authors,

I think you did not understand my comments. Actually, the structure of the article is still not suitable and the discussion part is still a result one. To be clearer, I never seen so many p values in a discussion for example. A discussion discusses the results (for exemple the p values presented in the result part), and does not make a new list of the articles you included. Is not necessary long but it involves your own comments on the articles included based on evidences or methodological discussion or comparisons or hypotheses and so on. On the other hand, you have not tried to be more precise in the PANS definition, you have not tried to include other case reports (a case report is not an evidence) supporting your introduction (PANS is different from COVID-19, we coud imagine COVID-19 is involved in PANS but there are also other pathogens involved in PANS and more clearly identified) etc etc...

Author Response

Reviewer 2

Dear revisor, in order to improve the paper we have added the corrections indicated, aware of the main limitations of the study, but which, as defined by the Academic Publisher, are inherent in the subject matter. Further evidence is needed to clarify the pathology, but our work could serve as a starting point for future perspectives. I hope you appreciate your efforts. Yours sincerely.

Comment: I think you did not understand my comments. Actually, the structure of the article is still not suitable and the discussion part is still a result one.

  • Response: Dear revisor, thanks for the opinion. We appreciated your suggestion to improve the paper. However, regarding the structure of the paper we followed the international guidelines set out by the PRISMA statement and PICOS framework as described and cited in the methods and results. The structure of the paper therefore cannot be modified because it must comply with the regulations of the guidelines.

Comment: To be clearer, I never seen so many p values in a discussion for example. A discussion discusses the results (for exemple the p values presented in the result part), and does not make a new list of the articles you included.

  • Response: Dear revisor, in order to avoid lists of numerical values, we have introduced in the text explanations of each facet of the included studies, analyzing the concepts and only then reporting the correlated numbers of the values.

Comment: Is not necessary long but it involves your own comments on the articles included based on evidences or methodological discussion or comparisons or hypotheses and so on.

  • Response Dear revisor, as suggested I’ve added several sentences that discussed the data or evidences present in the literature, comparing them with previous or our metanalysis, and proposing hypotheses. Moreover we emphasized what was also said by the academic editor on the limits of the reported evidence, critically analyzing the structural limits of the recovered papers as below:

‘’ However, the evidence that analyzes the efficacy of surgical treatment remains scarce, especially in the differentiation of outcomes based on the type of intervention admin-istered (tonsillectomy, adenoidectomy or adenotonsillectomy).Furthermore, the results of the surgical treatment should be compared with an adequate control group composed of a homogeneous sample treated with a medical therapy validated in the literature. Prasad et al. attempted to compare the outcomes achieved through surgical treatment alone versus those obtained from the combined surgical approach with intravenous immunoglobulins [20]. However, the authors enrolled a small group of patients, di-viding them into three different treatment arms: tonsillectomy and adenoidectomy (AT) (n = 28), AT plus intravenous immunoglobulin (IVIG) (n = 22), or non-surgical treatment (n = 10). Although caregivers did not report a decrease in symptom frequency depend-ing on the type of treatment except choreiform movement (p = 0.0296), TA was shown to be the treatment with the greatest symptom impact for patients (p = 0.05).  Another limitation of the study was that it did not administer only the treatments de-scribed in the three study arms. These patients benefited from further treatments such as antibiotics (n = 60, 100%), Rituximab (15%), steroids (20%), and plasma exchange (10%) which constituted a potential risk of bias. Another issue of concern remains the effectiveness of surgical treatment on disease prevention by influencing the onset of neuropsychiatric symptoms. Nevertheless, when examining the literature, the only quantifiable outcome remains the change in OCD before and after treatment as reported in our systematic review, not allowing a quan-titative analysis of the results and, therefore, not drawing valid conclusions.

  • Response: Moreover, also in the following periods of the discussion we have paid attention to the limits of each study, modifying the concepts to make the text more fluent and we have also hypothesized the reason for the limitations of each study or the possible beneficial effect for achieving excellent post-treatment outcomes as below:
  • We interpreted these data interesting because although statistical significance has not been achieved, this may be simply due to the lack of evidence in the literature on this subject, particularly insufficient sample sizes, study protocols, and non-standardized selection criteria that do not allow adequate comparisons.
  • Murphy et al. in 2002 he reported promising results of medical therapy, finding a rapid resolution of the symptoms of OCD, anxiety and tics that occur after 14 days of appropriate antibiotic treatment in 6/12 patients with PANDAS [18]. The authors put forward an interesting point of view regarding the management of related symptoms. GABHS serological tests have been used as an objective evaluation of response to treatment and the eradication of the germ by antibiotics has shown efficacy in the resolution of OCD symptoms as well as any relapse after acute streptococcal infection. Therefore, the authors consistently with what was hypothesized
  • in the study, they obtained the predetermined outcomes, even though they enrolled an insufficient sample of patients.
  • However, it should be noted that the authors in this case did not correctly correlate the number of exacerbation and the eradication of the germ to the reduction of specific symptoms.
  • The lack of efficacy has probably been interpreted in the role of the eradication of the germ in the resolution of the pathology, regardless of the reduction of the episodes of inflammation

Comment: On the other hand, you have not tried to be more precise in the PANS definition, you have not tried to include other case reports (a case report is not an evidence) supporting your introduction (PANS is different from COVID-19, we coud imagine COVID-19 is involved in PANS but there are also other pathogens involved in PANS and more clearly identified) etc etc...

  • Response: Dear reviewer, as you suggested, we have clarified the difference between PANDAS and PANS diseases, adding further evidence to that previously reported. We share the relevant opinion of distinguishing diseases as they include several different etiological factors as below:

The clinical recommendations of the PANS Consensus Conference clarified the clinical evaluation and diagnostic criteria of patients with young pediatric Acute-Onset Neu-ropsychiatric Syndrome (PANS) [Chang et al.].

PANS describe a clinical defined by the sudden and dramatic onset of obses-sive-compulsive disorder or severely restricted food intake, associated with acute onsets neuropsychiatric symptoms such as anxiety, emotional lability and/or depression, irri-tability, aggression, and/or strongly oppositional behavior behavioral regression (de-velopment), deterioration in school performance or memory impairment. Instead, the PANDAS subgroup is defined by an acute prepubertal onset of tics or OCD symptoms associated with GAS infection and specific neuropsychiatric symptoms. It is distin-guished from PANS by a sudden onset, and episodic course and tics have an "off / on" and increasing/decreasing course. While PANDAS has a specific infectious pathogen responsible, PANS foresee different microbes possibly implicated in the genesis of the disorders postinfectious neurological such as H1N1 influenza, Epstein Barr virus, and Borrelia burgdorferi (Lyme disease) [Ercan, Pavone, Fallon, Caruso].

and cited Wilbur C, Bitnun A, Kronenberg S, et al. PANDAS/PANS in childhood: Controversies and evidence. Paediatr Child Health. 2019;24(2):85-91. doi:10.1093/pch/pxy145, Chang K, Frankovich J, Cooperstock M, et al. Clinical evaluation of youth with pediatric acute-onset neuropsychiatric syndrome (PANS): recommendations from the 2013 PANS Consensus Conference. J Child Adolesc Psychopharmacol. 2015;25(1):3-13. doi:10.1089/cap.2014.0084

Wilbur C, Bitnun A, Kronenberg S, et al. PANDAS/PANS in childhood: Controversies and evidence. Paediatr Child Health. 2019;24(2):85-91. doi:10.1093/pch/pxy145,

Ercan TE, Ercan G, Severge B, Arpaozu M, Karasu G. Mycoplasma pneumoniae infection and obsessive-compulsive disease: a case report. J Child Neurol. 2008;23(3):338-340. doi:10.1177/0883073807308714

Caruso JM, Tung GA, Gascon GG, Rogg J, Davis L, Brown WD. Persistent preceding focal neurologic deficits in children with chronic Epstein-Barr virus encephalitis. J Child Neurol. 2000;15(12):791-796. doi:10.1177/088307380001501204

Fallon BA, Kochevar JM, Gaito A, Nields JA. The underdiagnosis of neuropsychiatric Lyme disease in children and adults. Psychiatr Clin North Am. 1998;21(3):693-viii. doi:10.1016/s0193-953x(05)70032-0

Best regards.

Round 3

Reviewer 2 Report

I have no more comment.